# WHAT CLASSIFIERS KNOW WHAT THEY DON'T KNOW?

## ABSTRACT

Being uncertain when facing the unknown is key to intelligent decision making. However, machine learning algorithms lack reliable estimates about their predictive uncertainty. This leads to wrong and overly-confident decisions when encountering classes unseen during training. Despite the importance of equipping classifiers with uncertainty estimates ready for the real world, most prior work focuses on small datasets with little or no class discrepancy between training and testing data. To close this gap, we introduce UIMNET: a realistic, ImageNet-scale test-bed to evaluate predictive uncertainty estimates for deep image classifiers. Our benchmark provides implementations of ten state-of-the-art algorithms, six uncertainty measures, four in-domain metrics, three out-domain metrics, and a fully automated pipeline to train, calibrate, ensemble, select, and evaluate models. Our test-bed is open-source and all of our results are reproducible from a fixed commit in our repository. Adding new datasets, algorithms, measures, or metrics is a matter of a few lines of code—in so hoping that UIMNET becomes a stepping stone towards realistic, rigorous, and reproducible research in uncertainty estimation. Our experimental results reveal that, in order to obtain the best possible uncertainty estimates in large-scale image classification, the practitioner should favor large, calibrated models. We recommend the use of single ERM models, single MIMO models, or ensembles of ERM models, in order of increasing performance and required computational budget.

## 1 INTRODUCTION

> *I don't think I've ever seen anything quite like this before*
> —HAL 9000 in *2001: A Space Odyssey*

Deep image classifiers exceed at discriminating the set of *in-domain* classes observed during training. However, when confronting test examples from unseen *out-domain* classes, these classifiers can only predict in terms of known categories, leading to wrong and overly-confident decisions (Hein et al., 2019; Ulmer & Cinà, 2020). In short, machine learning systems are unaware of their own limits, since "they do not know what they do not know". Since out-domain data cannot be safely identified and treated accordingly, it is reasonable to fear that, when deployed in-the-wild, the safety and performance of these classifiers crumbles by leaps and bounds (Ovadia et al., 2019). The inability of machine learning systems to estimate their uncertainty and abstaining to classify out-domain classes roadblocks their use in critical applications. These include self-driving (Michelmore et al., 2018), medicine (Begoli et al., 2019), and the analysis of satellite imagery (Wadoux, 2019). Good uncertainty estimates are also a key ingredient in anomaly detection (Chalapathy & Chawla, 2019), active learning (Settles, 2009), safe reinforcement learning (Henaff et al., 2019), defending against adversarial examples (Goodfellow et al., 2014), and model interpretability (Alvarez-Melis & Jaakkola, 2017). For an extensive literature review on uncertainty estimation and its applications, we refer the curious reader to the surveys of Abdar et al. (2020) and Ruff et al. (2021). Despite a research effort spanning multiple decades, machine learning systems still lack trustworthy estimates of their predictive uncertainty. In our view, one hindrance to this research program is the absence of realistic benchmarking and evaluation protocols. More specifically, prior attempts are limited in two fundamental ways. First, these experiment on small datasets such as SVHN and CIFAR-10 (van Amersfoort et al., 2021). Second, these do not provide a challenging set of out-domain data. Instead, they construct out-domain classes by using a second dataset (e.g., using MNIST in-domain versus FashionMNIST out-domain, cf. Van Amersfoort et al. (2020)) or by perturbing the in-domain classes

| Datasets | Algorithms | Uncertainty measures | In-Domain metrics | Out-Domain metrics | Ablations |
|---|---|---|---|---|---|
| ImageNot (new) | ERM Mixup Soft-labeler RBF RND OCs DeepAE MC-Dropout MIMO DUE (+ Ensembles) | Largest Gap Entropy Jacobian GMM Native | ACC@1 ACC@5 ECE NLL | AUC InAsIn OutAsOut | Calibration (y/n) Spectral norm (y/n) Model size (RN18 / RN50) |

Table 1: The UIMNET test-bed suite for uncertainty estimation.

using handcrafted transformations (such as Gaussian noise or blur, see ImageNet-C Hendrycks & Dietterich (2019)). Both approaches result in simplistic benchmarking, and little is learned about uncertainty estimation for the real world. One exception is the ImageNet-O dataset (Hendrycks et al., 2021), where out-domain data is selected from ImageNet-22k classes not contained in ImageNet-1k. However, the images in ImageNet-O are collected adversarially as to maximize the prediction confidence of ResNet50 ImageNet-1k classifiers, creating what's possibly an unnecessarily difficult benchmark. The purpose of this work is to introduce an end-to-end benchmark and evaluation protocol as realistic as possible. At the time of writing, UIMNET is the most exhaustive benchmark for uncertainty estimation in the literature.

**Formal setup**  We learn classifiers $f$ using *in-domain* data from some distribution $P_{\text{in}}(X, Y)$. After training, we endow the classifier with a real-valued uncertainty measure $u(f, x^\dagger)$. Given a test example $(x^\dagger, y^\dagger) \sim P$ with unobserved label $y^\dagger$, we declare $x^\dagger$ *in-domain* (hypothesizing $P = P_{\text{in}}$) if $u(f, x^\dagger)$ is small, whereas we declare $x^\dagger$ *out-domain* (hypothesizing $P \neq P_{\text{in}}$) if $u(f, x^\dagger)$ is large. Using these tools, our goal is to abstain from classifying out-domain test examples, and to classify with calibrated probabilities in-domain test examples. The sequel assumes that the difference between in- and out-domain resides in that the two groups of data concern disjoint classes.

**Contributions**  We introduce UIMNET, a test-bed for large-scale, realistic evaluation of uncertainty estimates in deep image classifiers. We build UIMNET as follows (see also Table 1).

(Sec. 2) We construct ImageNot, a perceptual partition of ImageNet into *in-domain* and *out-domain* classes. Unlike most prior work focusing on small datasets like SVHN and CIFAR-10, ImageNot provides a benchmark for uncertainty estimators at a much larger scale. Moreover, both in-domain and out-domain categories in ImageNot originate from the original ImageNet dataset. This provides realistic out-domain data, as opposed to prior work relying on a second dataset (e.g., MNIST as in-domain versus SVHN as out-domain), or handcrafted perturbations of in-domain classes (Gaussian noise or blur as out-domain).

(Sec. 3) We re-implement eight state-of-the-art algorithms from scratch, listed in Table 1. This allows a fair comparison under the exact same experimental conditions (training/validation splits, hyper-parameter search, neural network architectures and random initializations). Furthermore, we also study ensembles of multiple training instances for each algorithm.

(Sec. 4) Each algorithm can be endowed with one out of six possible uncertainty measures, allowing an exhaustive study of what algorithms play well with what measures. Listed in Table 1, these are the largest softmax score, the gap between the two largest softmax scores, the softmax entropy, the norm of the Jacobian, a per-class Gaussian density model, and (for those available) an algorithm-specific measure.

(Sec. 5) For each classifier-measure pair, we study four in-domain metrics (top-1 and top-5 classification accuracy, log-likelihood, expected calibration error) and three out-domain metrics (the AUC at classifying in-domain versus out-domain samples using the selected

uncertainty measure, as well as the confusion matrix at a fixed uncertainty threshold computed over an in-domain validation set).

(Sec. 6) We explore three popular ablations to understand the impact of model calibration by temperature scaling, model size, and the use of spectral normalization.

(Sec. 7) UIMNET is entirely hands-off, since the pipeline from zero to LaTeX tables is fully automated: this includes hyper-parameter search, model calibration, model ensembling, and the production of all the tables included in our experimental results.

(Sec. 8) Our experimental results reveal that, in order to obtain the best possible uncertainty estimates in large-scale image classification, the practitioner should favor large, calibrated models. We recommend the use of single ERM models, single MIMO models, or ensembles of ERM models, in order of increasing performance and required computational budget.

UIMNET is open sourced at `https://github.com/ANONYMOUS`. All of the tables presented in this paper are reproducible by running the main script in the repository at commit `0xANON`.

## 2 CONSTRUCTING THE IMAGENOT BENCHMARK

The ImageNet dataset (Russakovsky et al., 2015) is a gold standard to conduct research in computer vision pertaining image data of 1000 different classes. Here we use the ImageNet dataset to derive ImageNot, a large-scale and realistic benchmark for uncertainty estimation. ImageNot partitions the 1000 classes of the original ImageNet dataset into *in-domain* classes (used to train and evaluate algorithms in-distribution) and *out-domain* classes (used to evaluate algorithms out-of-distribution).

To partition ImageNet into in-domain and out-domain, we featurize the entire dataset to understand the perceptual similarity between classes. To this end, we use a pre-trained ResNet-18 (He et al., 2016) to compute the average last-layer representation for each of the classes. Next, we use agglomerative hierarchical clustering Ward Jr (1963) to construct a tree describing the perceptual similarities between the 1000 average feature vectors. Such perceptual tree has 1000 leafs, each of them being a cluster containing one of the classes. During each step of the iterative agglomerative clustering algorithm the two closest clusters are merged, where the distance between two clusters is computed using the criterion of Ward Jr (1963). The algorithm halts when there are only two clusters left to merge, forming the root node of the tree.

At this point, we declare the 266 classes to the left of the root as *in-domain*, and the first 266 classes to the right of the root as *out-domain*. In the sequel, we call "training set" and "validation set" to a 90/10 random split from the original ImageNet "train" set. We call "testing set" to the original ImageNet "val" split. The exact in-/out-domain class partition as well as the considered train/validation splits are specified in Appendix D. Our agglomerative clustering procedure ended-up congregating different types of objects as in-domain classes, while grouping animals as out-domain classes.

While inspired by the BREEDS ImageNet splits (Santurkar et al., 2020), our benchmark ImageNot is conceived to tackle a different problem. The aim of the BREEDS dataset is to classify ImageNet into a small number of super-classes, each of them containing a number of perceptually-similar sub-classes. The BREEDS training and testing distributions differ on the sub-class proportions contributing to their super-classes. Since the BREEDS task is to classify super-classes, the set of labels remains constant from training to testing conditions. This is in contrast to ImageNot, where the algorithm observes only in-domain classes during training, but both in-domain and out-domain classes during evaluation. While BREEDS studies the important problem of domain generalization (Gulrajani & Lopez-Paz, 2020), where there is always a right prediction to make within the in-domain classes during evaluation, here we focus on measuring uncertainty and abstaining from predicting about those out-domain classes unavailable during training.

ImageNot is also similar to the ImageNet-O dataset of (Hendrycks et al., 2021). However, their out-domain images are collected adversarially, that is, to maximize the prediction confidence of ResNet50 classifiers. We believe that this drastic change in selection bias from ImageNet to ImageNet-O may result in an unnecessarily difficult uncertainty estimation benchmark. Thus, here we favor using only the original ImageNet data, as described above. In contrast, the starting point for both in-domain and

out-domain classes of our ImageNot is the ImageNet dataset, and thus should maximally overlap in terms of image statistics, leading to a challenging and realistic benchmark.

## 3  ALGORITHMS

We benchmark ten supervised learning algorithms commonly applied to tasks involving uncertainty estimation. Each algorithm consumes one in-domain training set of image-label pairs $\{(x_i, y_i)\}_{i=1}^n$ and returns a *predictor* $f(x) = w(\phi(x))$, composed by a *featurizer* $\phi : \mathbb{R}^{3 \times 224 \times 224} \to \mathbb{R}^k$ and a *classifier* $w : \mathbb{R}^k \to \mathbb{R}^C$. We consider predictors implemented using deep convolutional neural networks (LeCun et al., 2015). Given an input image $x^\dagger$, all predictors return a softmax vector $f(x^\dagger) = (f(x^\dagger)_c)_{c=1}^C$ over $C$ classes. The considered algorithms are:

- **Empirical Risk Minimization**, or ERM (Vapnik, 1992), or vanilla training.
- **Mixup** (Zhang et al., 2017) chooses a predictor minimizing the empirical risk on *mixed* examples $(\lambda \cdot x_i + (1 - \lambda) \cdot x_j, \lambda \cdot y_i + (1 - \lambda) \cdot y_j)$, where $\lambda \sim \text{Beta}(\alpha, \alpha)$, $\alpha$ is a mixing parameter, and $((x_i, y_i), (x_j, y_j))$ is a random pair of training examples. Mixup improves generalization performance (Zhang et al., 2017) and calibration (Thulasidasan et al., 2019).
- **Random Network Distillation**, or RND (Burda et al., 2018), finds an ERM predictor $f(x) = w(\phi(x))$, but simultaneously trains an auxiliary classifier $w_{\text{student}}$ to minimize $\|w_{\text{student}}(\phi(x)) - w_{\text{teacher}}(\phi(x))\|_2^2$, where $w_{\text{teacher}}$ is a fixed classifier with random weights. RND has shown good performance as a tool for exploration in reinforcement learning.
- **Orthogonal Certificates**, or OC (Tagasovska & Lopez-Paz, 2018), is analogous to RND for $w_{\text{teacher}}(\phi(x)) = \vec{0}_k$ for all $x$. That is, the goal of $w_{\text{student}}$ is to map all the in-domain training examples to zero in $k$ different ways (or *certificates*). To ensure diverse and non-trivial certificates, we regularize each weight matrix $W$ of $w_{\text{student}}$ to be orthogonal by adding a regularization term $\|W^\top W - I\|_2^2$. OCs have shown good performance at the task of estimating uncertainty across a variety of classification tasks.
- **Autoencoder**, or DeepAE (Vincent et al., 2010), is analogous to RND for $w_{\text{teacher}}(\phi(x)) = \phi(x)$ and a $w_{\text{student}}$ with a bottleneck.
- **MC-Dropout** (Gal & Ghahramani, 2016) trains ERMs with one or more dropout layers (Srivastava et al., 2014). These stochastic dropout layers remain active at test time, allowing the predictor to produce multiple softmax vectors $\{f(x^\dagger, \text{dropout}_t)\}_{t=1}^T$ for each test example $x^\dagger$. Here, $\text{dropout}_t$ is a random dropout mask sampled anew. MCDropout is one of the most popular baselines to estimate uncertainty.
- **MIMO** (Havasi et al., 2021) is a variant of ERM over predictors accepting $T$ images and producing $T$ softmax vectors. For example, MIMO with $T = 3$ is trained to predict jointly the label vector $(y_i, y_j, y_k)$ using a predictor $h(x_i, x_j, x_k)$, where $((x_t, y_t))_{i=1}^3$ is a random triplet of training examples. Given a test point $x^\dagger$, we form predictions by replicating and averaging, that is $f(x^\dagger) = \frac{1}{3} \sum_{t=1}^3 h(x^\dagger, x^\dagger, x^\dagger)_t$.
- **Radial Basis Function**, or RBF (Broomhead & Lowe, 1988), is a variant of ERM where we transform the logit vector $z \mapsto e^{-z^2}$ before passing them to the final softmax layer. In such a way, as the logit norm $\|z\| \to \infty$, the predicted softmax vector tends to the maximum entropy solution $(\frac{1}{C})_{c=1}^C$, signaling high uncertainty far away from the training data. RBFs have been proposed as defense to adversarial examples (Goodfellow et al., 2014), but they remain under-explored given the difficulties involved in their training.
- **Soft labeler** (Hinton et al., 2015; Szegedy et al., 2016) is a variant of ERM where the one-hot vector labels $y_i$ are *smoothed* such that every zero becomes $\ell_{\min} > 0$ and the solitary one becomes $\ell_{\max} < 1$. Using soft labels, we can identify softmax vectors with entries exceeding $\ell_{\max}$ as "over-shoots", and regard them as uncertain predictions.
- **DUE** (van Amersfoort et al., 2021) enforces the smoothness of the featurizer $\phi$ using spectral normalization (Miyato et al., 2018), implements the classifier $w$ as a sparse Gaussian Process (Quinonero-Candela & Rasmussen, 2005), and trains the resulting predictor using variational inference (Titsias, 2009). Gaussian processes are considered one of the main tools to estimate predictive uncertainty in machine learning systems.

**Ensembles of predictors** We also consider ensembles of predictors trained by each of the algorithms above. Ensembles are commonly regarded as the state-of-the-art in uncertainty estimation (Lakshminarayanan et al., 2016). In particular, and for each algorithm, we construct bagging ensembles

by (i) selecting the best $K \in \{1, 5\}$ predictors $\{f^k\}_{k=1}^K$ from all considered random initializations and hyper-parameters, and (ii) returning the average function $f(x^\dagger) := \frac{1}{K} \sum_{k=1}^M f^k(x^\dagger)$.

## 4 Uncertainty Measures

We equip a trained predictor $f$ with six different uncertainty measures. An uncertainty measure is a real-valued function $u(f, x^\dagger)$ designed to return small values for *in-domain* instances $x^\dagger$, and large values for *out-domain* instances $x^\dagger$. To describe the different measures, let $\{s_{(1)}, \ldots, s_{(C)}\}$ be the softmax scores returned by $f(x^\dagger)$ sorted in decreasing order.

- **Largest** (Hendrycks & Gimpel, 2016) returns minus the largest softmax score, $-s_{(1)}$
- **Softmax gap** (Tagasovska & Lopez-Paz, 2018) returns $s_{(2)} - s_{(1)}$.
- **Entropy** (Shannon, 1948) returns $-\sum_{c=1}^C s_{(c)} \mathbb{I}\{s_{(c)} > 0\} \log s_{(c)}$.
- **Norm of the Jacobian** (Novak et al., 2018) returns $\|\nabla_x f(x^\dagger)\|_2^2$.
- **GMM** (Mukhoti et al., 2021) estimates one Gaussian density $\mathcal{N}(\phi(x); \mu_c, \Sigma_c)$ per-class, on top of the feature vectors $\phi(x)$ collected from a in-domain validation set. Given a test example $x^\dagger$, return $-\sum_{c=1}^C \lambda_c \cdot \mathcal{N}(\phi(x^\dagger); \mu_c, \Sigma_c)$, where $\lambda_c$ is the proportion of in-domain validation examples from class $c$.
- **Test-time augmentation** (Ashukha et al., 2020) returns $-\max_c(\frac{1}{A} \sum_{a=1}^A f(x_a^\dagger))_c$. This is the measure "**Largest**" about the average prediction over $A$ random data augmentations $\{x_a^\dagger\}_{a=1}^A$ of the test instance $x^\dagger$.

These uncertainty measures are applicable to all the algorithms considered in Section 3. Additionally, some algorithms provide their **Native** uncertainty measures, outlined below.

- For **Mixup**, we return $\frac{1}{K} \sum_{k=1}^K \|\lambda \cdot f(x^\dagger) + (1 - \lambda) \cdot \bar{y}_k - f(\lambda \cdot x^\dagger + (1 - \lambda) \cdot \bar{x}_k)\|_2^2$, where $\lambda \sim \text{Beta}(\alpha, \alpha)$, and $(\bar{x}_k, \bar{y}_k)$ is an example saved from the training set. This measures if the test example $x^\dagger$ violates the Mixup criterion wrt the training dataset average.
- For **RND**, **OC**, and **DeepAE** we return $\|w_{\text{student}}(\phi(x^\dagger)) - w_{\text{teacher}}(\phi(x^\dagger))\|_2^2$, that is, we consider a prediction uncertain if the outputs of the student and teacher disagree. We expect this disagreement to be related predictive uncertainty, as the student did not observe the behaviour of the teacher at out-domain instances $x^\dagger$.
- **For Soft labeler** we return $(s_{(1)} - \ell_{\max})^2$. This measures the discrepancy between the largest softmax and the positive soft label target, able to signal overly-confident predictions.
- For **MC-Dropout** and **Ensembles**, and following (Lakshminarayanan et al., 2016), we return the Jensen-Shannon divergence between the $K$ members (or stochastic forward passes) $f^1, \ldots, f^K$ of the ensemble, $H\left(\frac{1}{K} \sum_{k=1}^K f^k(x^\dagger)\right) - \frac{1}{K} \sum_{k=1}^K H(f^k(x^\dagger))$.
- For **DUE**, we return the predictive variance of the Gaussian process classifier.

## 5 Evaluation Metrics

For each algorithm-measure pair, we evaluate several metrics both *in-domain* and *out-domain*.

Following (Havasi et al., 2021), we implement four **in-domain metrics** to assess the performance and calibration of predictors when facing in-domain test examples.

- **Top-1** and **Top-5** classification accuracy (Russakovsky et al., 2015).
- **Expected Calibration Error** or ECE (Guo et al., 2017): $\frac{1}{B} \sum_{b=1}^B \frac{|B_b|}{n} |\text{acc}(f, B_b) - \text{conf}(f, B_b)|$, where $B_b$ contains the examples where the algorithm predicts a softmax score of $b$. The functions *acc* and *conf* compute the average classification accuracy and largest softmax score of $f$ over $B_b$. ECE is minimized when $f$ is calibrated, that is, $f$ is wrong $p\%$ of the times it predicts a largest softmax score $p$. Following (Guo et al., 2017), we discretize $b \in [0, 1]$ into 15 equally-spaced bins.
- **Negative Log Likelihood (NLL)** Also known as the cross-entropy loss, this is the objective minimized during the training process of the algorithms.

We assess the uncertainty estimates of each predictor-measure pair using three **out-domain metrics**.

- **Area Under the Curve**, or AUC (Tagasovska & Lopez-Paz, 2018), describes how well does the predictor-measure pair distinguish between in-domain and out-domain examples over all thresholds of the uncertainty measure.
- **Confusion matrix at fixed threshold**. To reject out-domain examples in real scenarios, one must fix a threshold $\theta$ for the selected uncertainty measure. We do so by computing the 95% quantile of the uncertainty measure, computed over an in-domain validation set. Then, at testing time, we declare one example out-domain if the uncertainty measure exceeds $\theta$.[1] To understand where does the uncertainty measure hit or miss, we monitor the metrics **InAsIn** (percentage of in-domain examples classified as in-domain) and **OutAsOut** (percentage of in-domain examples classified as out-domain).

## 6 ABLATIONS

We execute our entire test-bed under three popular ablations.

- We study the effect of **calibration** by temperature scaling (Platt et al., 1999). To this end, we introduce a temperature scaling $\tau > 0$ before the softmax layer, resulting in predictions Softmax($\frac{z}{\tau}$) about the logit vector $z$. We estimate the optimal temperature $\hat{\tau}$ by minimizing the **NLL** of the predictor across an in-domain validation set. We evaluate all metrics for both the un-calibrated ($\tau = 1$) and calibrated ($\tau = \hat{\tau}$) predictors. According to previous literature (Guo et al., 2017), calibrated models provide better in-domain uncertainty estimates.
- We analyze the impact of **spectral normalization** to control the behavior of the featurizer $\phi$. More specifically, recent works (Liu et al., 2020; Van Amersfoort et al., 2020; van Amersfoort et al., 2021; Mukhoti et al., 2021) have highlighted the importance of controlling both the *smoothness* and *sensitivity* of the feature extraction process to avoid *feature collapse* and achieve high-quality uncertainty estimates. On the one hand, enforcing *smoothness upper bounds* the Lipschitz constant of $\phi$, limiting its reaction to changes in the input. Smoothness is often enforced by normalizing each weight matrix in $\phi$ by its spectral norm (Miyato et al., 2018). On the other hand, enforcing *sensitivity lower bounds* the Lipschitz constant of $\phi$, ensuring that the feature space reacts in some amount when the input changes. Sensitivity is often enforced by the residual connections of the hereby used ResNet models (He et al., 2016). We apply one-sided spectral normalization, with a target spectral norm of 5, to the weights and the batch normalization layers following Miyato et al. (2018) and Gouk et al. (2021) respectively.
- We analyze the impact of the model size (ResNet-18 versus ResNet-50).

## 7 EXPERIMENTAL PROTOCOL

We now conduct experiments on the ImageNot benchmark (Section 2) for all algorithms (Section 3) and measures (Section 4), wrt all metrics (Section 5) and ablations (Section 6). *Empirical oracle upper bounds* To have a measure of the maximally achievable separation of the in-domain and out-of-domain partitions, we train a ResNet-18 classifier to discriminate between the In-domain classes and the out-of-domain partition. The oracle performance is $0.985 \pm 0.002$, $0.944 \pm 0.001$ and $0.947 \pm 0.002$ for the AUC, InAsIn and OutAsOut metrics respectively.

*Hyper-parameter search.* We train each algorithm using (i) ResNet-18 or ResNet-50 architectures (ii) spectral normalization or not, (iii) ten hyper-parameter trials, and (iv) three random train/validation splits of the in-domain data (data seeds). We opt for a random hyper-parameter search (Bergstra & Bengio, 2012), where the search grid for each algorithm is detailed in Appendix C. More specifically, while the first trial uses the default hyper-parameter configuration suggested by the authors of each algorithm, the additional four trials explore random hyper-parameters.

*Model selection.* After training all instances of a given algorithm, we report the *test* average and standard deviation (over data seeds) of all metrics. We report these metrics for (a) the best model

---

[1]This strategy is equivalent to the statistical hypothesis test with null "$H_0$: the observed example is in-domain".

($k = 1$), and (b) the ensemble containing the best five models ($k = 5$) in terms of the *validation* average (over data seeds) negative log-likelihood[2].

*Optimization.*    We use PyTorch (Paszke et al., 2019) and SGD (Bottou, 2012) for 100 epochs using mini-batches of 256 examples distributed over 8 NVIDIA Tesla V100 GPUs, and a learning rate decaying by a factor of 10 every 30 epochs.

## 8    RESULTS

The full experimental results are available in Appendix A (in-domain) and Appendix B (out-domain). Below, we summarize our findings.

From the **in-domain** results summarized in Table 2, we identify the following key takeaways:

- No single method out-performs ERM significantly on any in-domain metric, except MIMO on ECE (-30% ECE).
- No ensemble method outperforms ensembles of ERMs in any in-domain metric.
- Increasing model size is the most effective strategy to improve in-domain metrics (+5% ACC@1, +2% ACC@5, -17% NLL, -13% ECE).
- Ensembling is the second most effective strategy to improve in-domain metrics (+2.5% ACC@1, +1% ACC@5, -8.5% NLL, -8% ECE).
- Calibration is an effective strategy to improve test negative log-likelihood and expected calibration error (-2% NLL, -45% ECE).
- Spectral normalization has little effect to improve in-domain metrics (<1%).
- DUE has a brittle behavior, performing well only for default hyper-parameters and ResNet18 architectures.

From the **out-domain** results summarized in Table 3, corresponding to the best performing uncertainty measure "Entropy", we identify the following key takeaways:

- No single method outperforms ERM significantly on any out-domain metric, except MIMO on OutAsOut (+5% OutAsOut).
- No ensemble method outperforms ensembles of ERMs in any out-domain metric.
- Increasing model size is the most effective strategy to improve out-domain metrics (+4% AUC, +35% OutAsOut).
- Ensembling is the second most effective strategy to improve out-domain metrics (+1% AUC, +4% OutAsOut).
- Calibration has a small negative effect on out-domain metrics (-1%).
- Spectral normalization has a small positive effect on out-domain metrics (<1%).
- All methods are able to upper-bound their type-I errors to the requested threshold of 5%.
- The best performing uncertainty measure is Entropy; then Largest, and Gap follow. The worst performing measures are Augmentations, Jacobian, and Native (Appendix B).

**Other results**    We were unable to obtain competitive performances for the algorithm RBF (Broomhead & Lowe, 1988) and the measure GMM (Mukhoti et al., 2021). We believe that training RBFs at this large scale are challenging optimization problems that deserve further study in our community. Furthermore, the large number of classes in our study (266 ImageNet classes instead of the 10 CIFAR-10 classes often considered) poses a difficult problem for the density-based measures GMM.

## 9    CONCLUSION

To obtain the best possible uncertainty estimates in large-scale image classification, we recommend the use of large and calibrated models. We favor single ERM models (low computational budget), single MIMO models (medium budget), or ensembles of ERM models (high budget).

---

[2]Graphs showing the relationship between validation set negative likelihood and out-of-domain measures on the test set are show in the appendix.

| algorithm | k | calibration | spectral | ACC@1 (↑) | ACC@5 (↑) | NLL (↓) | ECE (↓) |
|---|---|---|---|---|---|---|---|
| ERM | 1 | False | False | 0.767 ± 0.002 | 0.930 ± 0.000 | 0.939 ± 0.004 | 0.053 ± 0.001 |
| | | True | False | 0.767 ± 0.002 | 0.930 ± 0.000 | 0.918 ± 0.005 | 0.031 ± 0.000 |
| | | False | True | 0.766 ± 0.003 | 0.927 ± 0.001 | 0.947 ± 0.008 | 0.055 ± 0.002 |
| | | True | True | 0.766 ± 0.003 | 0.927 ± 0.001 | 0.924 ± 0.008 | 0.031 ± 0.002 |
| | 5 | False | False | 0.780 ± 0.007 | 0.933 ± 0.005 | 0.865 ± 0.008 | 0.030 ± 0.008 |
| | | True | False | 0.780 ± 0.007 | 0.933 ± 0.005 | 0.863 ± 0.029 | 0.021 ± 0.005 |
| | | False | True | 0.788 ± 0.001 | 0.937 ± 0.000 | 0.821 ± 0.003 | 0.019 ± 0.002 |
| | | True | True | 0.788 ± 0.001 | 0.937 ± 0.000 | 0.822 ± 0.003 | 0.019 ± 0.003 |
| Mixup | 1 | False | False | 0.768 ± 0.001 | 0.929 ± 0.000 | 0.925 ± 0.003 | 0.035 ± 0.001 |
| | | True | False | 0.768 ± 0.001 | 0.929 ± 0.000 | 0.923 ± 0.003 | 0.030 ± 0.002 |
| | | False | True | 0.766 ± 0.003 | 0.927 ± 0.001 | 0.942 ± 0.005 | 0.021 ± 0.004 |
| | | True | True | 0.766 ± 0.003 | 0.927 ± 0.001 | 0.940 ± 0.004 | 0.029 ± 0.001 |
| | 5 | False | False | 0.777 ± 0.001 | 0.933 ± 0.001 | 0.871 ± 0.002 | 0.039 ± 0.001 |
| | | True | False | 0.777 ± 0.001 | 0.933 ± 0.001 | 0.857 ± 0.002 | 0.017 ± 0.001 |
| | | False | True | 0.789 ± 0.000 | 0.940 ± 0.000 | 0.840 ± 0.001 | 0.051 ± 0.002 |
| | | True | True | 0.789 ± 0.000 | 0.940 ± 0.000 | 0.814 ± 0.002 | 0.019 ± 0.002 |
| SoftLabeler | 1 | False | False | 0.767 ± 0.003 | 0.930 ± 0.001 | 0.978 ± 0.009 | 0.032 ± 0.001 |
| | | True | False | 0.767 ± 0.003 | 0.930 ± 0.001 | 0.949 ± 0.010 | 0.043 ± 0.003 |
| | | False | True | 0.767 ± 0.001 | 0.928 ± 0.001 | 0.984 ± 0.003 | 0.030 ± 0.002 |
| | | True | True | 0.767 ± 0.001 | 0.928 ± 0.001 | 0.958 ± 0.006 | 0.045 ± 0.002 |
| | 5 | False | False | 0.782 ± 0.006 | 0.935 ± 0.003 | 1.013 ± 0.073 | 0.148 ± 0.040 |
| | | True | False | 0.782 ± 0.006 | 0.935 ± 0.003 | 0.861 ± 0.025 | 0.029 ± 0.003 |
| | | False | True | 0.790 ± 0.001 | 0.940 ± 0.001 | 0.906 ± 0.001 | 0.089 ± 0.003 |
| | | True | True | 0.790 ± 0.001 | 0.940 ± 0.001 | 0.827 ± 0.002 | 0.032 ± 0.001 |
| DeepAE | 1 | False | False | 0.765 ± 0.001 | 0.928 ± 0.001 | 0.933 ± 0.004 | 0.046 ± 0.003 |
| | | True | False | 0.765 ± 0.001 | 0.928 ± 0.001 | 0.922 ± 0.003 | 0.031 ± 0.001 |
| | | False | True | 0.765 ± 0.001 | 0.928 ± 0.001 | 0.934 ± 0.003 | 0.046 ± 0.003 |
| | | True | True | 0.765 ± 0.001 | 0.928 ± 0.001 | 0.923 ± 0.003 | 0.030 ± 0.002 |
| | 5 | False | False | 0.778 ± 0.008 | 0.934 ± 0.004 | 0.862 ± 0.036 | 0.032 ± 0.009 |
| | | True | False | 0.778 ± 0.008 | 0.934 ± 0.004 | 0.855 ± 0.031 | 0.015 ± 0.001 |
| | | False | True | 0.789 ± 0.001 | 0.940 ± 0.000 | 0.811 ± 0.001 | 0.020 ± 0.000 |
| | | True | True | 0.789 ± 0.001 | 0.940 ± 0.000 | 0.812 ± 0.001 | 0.015 ± 0.002 |
| RND | 1 | False | False | 0.762 ± 0.001 | 0.927 ± 0.001 | 0.957 ± 0.008 | 0.056 ± 0.001 |
| | | True | False | 0.762 ± 0.001 | 0.927 ± 0.001 | 0.936 ± 0.007 | 0.032 ± 0.002 |
| | | False | True | 0.764 ± 0.002 | 0.926 ± 0.001 | 0.955 ± 0.008 | 0.055 ± 0.002 |
| | | True | True | 0.764 ± 0.002 | 0.926 ± 0.001 | 0.933 ± 0.007 | 0.029 ± 0.001 |
| | 5 | False | False | 0.781 ± 0.007 | 0.936 ± 0.004 | 0.846 ± 0.031 | 0.025 ± 0.008 |
| | | True | False | 0.781 ± 0.007 | 0.936 ± 0.004 | 0.846 ± 0.027 | 0.016 ± 0.001 |
| | | False | True | 0.788 ± 0.001 | 0.938 ± 0.001 | 0.824 ± 0.002 | 0.019 ± 0.002 |
| | | True | True | 0.788 ± 0.001 | 0.938 ± 0.001 | 0.825 ± 0.002 | 0.018 ± 0.001 |
| OC | 1 | False | False | 0.764 ± 0.001 | 0.928 ± 0.001 | 0.947 ± 0.002 | 0.054 ± 0.001 |
| | | True | False | 0.764 ± 0.001 | 0.928 ± 0.001 | 0.926 ± 0.002 | 0.032 ± 0.001 |
| | | False | True | 0.765 ± 0.002 | 0.927 ± 0.002 | 0.948 ± 0.002 | 0.054 ± 0.001 |
| | | True | True | 0.765 ± 0.002 | 0.927 ± 0.002 | 0.927 ± 0.002 | 0.029 ± 0.002 |
| | 5 | False | False | 0.782 ± 0.008 | 0.935 ± 0.005 | 0.844 ± 0.034 | 0.022 ± 0.008 |
| | | True | False | 0.782 ± 0.008 | 0.935 ± 0.005 | 0.843 ± 0.031 | 0.016 ± 0.002 |
| | | False | True | 0.788 ± 0.004 | 0.937 ± 0.000 | 0.824 ± 0.002 | 0.020 ± 0.002 |
| | | True | True | 0.788 ± 0.004 | 0.937 ± 0.000 | 0.825 ± 0.002 | 0.018 ± 0.002 |
| MIMO | 1 | False | False | 0.768 ± 0.002 | 0.929 ± 0.000 | 0.929 ± 0.001 | 0.048 ± 0.001 |
| | | True | False | 0.768 ± 0.002 | 0.929 ± 0.000 | 0.906 ± 0.001 | 0.022 ± 0.001 |
| | | False | True | 0.768 ± 0.002 | 0.928 ± 0.001 | 0.930 ± 0.004 | 0.051 ± 0.001 |
| | | True | True | 0.768 ± 0.002 | 0.928 ± 0.001 | 0.905 ± 0.004 | 0.023 ± 0.000 |
| | 5 | False | False | 0.769 ± 0.008 | 0.928 ± 0.005 | 0.918 ± 0.042 | 0.048 ± 0.012 |
| | | True | False | 0.769 ± 0.008 | 0.928 ± 0.005 | 0.905 ± 0.033 | 0.015 ± 0.001 |
| | | False | True | 0.779 ± 0.001 | 0.934 ± 0.001 | 0.859 ± 0.004 | 0.032 ± 0.003 |
| | | True | True | 0.779 ± 0.001 | 0.934 ± 0.001 | 0.857 ± 0.004 | 0.018 ± 0.002 |
| MCDropout | 1 | False | False | 0.766 ± 0.002 | 0.928 ± 0.001 | 0.944 ± 0.002 | 0.054 ± 0.001 |
| | | True | False | 0.766 ± 0.002 | 0.928 ± 0.001 | 0.923 ± 0.001 | 0.031 ± 0.002 |
| | | False | True | 0.765 ± 0.002 | 0.927 ± 0.001 | 0.950 ± 0.003 | 0.055 ± 0.002 |
| | | True | True | 0.765 ± 0.002 | 0.927 ± 0.001 | 0.927 ± 0.002 | 0.030 ± 0.001 |
| | 5 | False | False | 0.777 ± 0.008 | 0.934 ± 0.004 | 0.867 ± 0.032 | 0.029 ± 0.007 |
| | | True | False | 0.777 ± 0.008 | 0.934 ± 0.004 | 0.865 ± 0.030 | 0.015 ± 0.003 |
| | | False | True | 0.788 ± 0.002 | 0.938 ± 0.001 | 0.822 ± 0.001 | 0.020 ± 0.001 |
| | | True | True | 0.788 ± 0.002 | 0.938 ± 0.001 | 0.823 ± 0.001 | 0.018 ± 0.001 |
| DUE | 1 | False | False | 0.086 ± 0.002 | 0.246 ± 0.005 | 4.548 ± 0.016 | 0.032 ± 0.003 |
| | | True | False | 0.086 ± 0.003 | 0.245 ± 0.005 | 4.536 ± 0.019 | 0.010 ± 0.002 |
| | | False | True | 0.653 ± 0.000 | 0.830 ± 0.000 | 1.575 ± 0.000 | 0.068 ± 0.000 |
| | | True | True | 0.654 ± 0.000 | 0.830 ± 0.000 | 1.569 ± 0.000 | 0.023 ± 0.000 |

Table 2: In-domain results for ResNet50.

| algorithm | k | calibration | spectral | AUC (↑) | InAsIn (↑) | OutAsOut (↑) |
|---|---|---|---|---|---|---|
| ERM | 1 | False | False | $0.865 \pm 0.003$ | $0.942 \pm 0.001$ | $0.373 \pm 0.006$ |
| | | False | False | $0.871 \pm 0.004$ | $0.943 \pm 0.001$ | $0.377 \pm 0.007$ |
| | | True | True | $0.864 \pm 0.003$ | $0.944 \pm 0.001$ | $0.368 \pm 0.012$ |
| | | True | True | $0.871 \pm 0.003$ | $0.944 \pm 0.001$ | $0.374 \pm 0.010$ |
| | 5 | False | False | $0.878 \pm 0.002$ | $0.943 \pm 0.002$ | $0.397 \pm 0.003$ |
| | | False | False | $0.876 \pm 0.002$ | $0.943 \pm 0.002$ | $0.397 \pm 0.002$ |
| | | True | True | $0.878 \pm 0.001$ | $0.942 \pm 0.001$ | $0.398 \pm 0.003$ |
| | | True | True | $0.877 \pm 0.001$ | $0.942 \pm 0.001$ | $0.397 \pm 0.003$ |
| Mixup | 1 | False | False | $0.858 \pm 0.002$ | $0.943 \pm 0.000$ | $0.353 \pm 0.004$ |
| | | False | False | $0.858 \pm 0.002$ | $0.943 \pm 0.000$ | $0.353 \pm 0.005$ |
| | | True | True | $0.855 \pm 0.007$ | $0.940 \pm 0.001$ | $0.348 \pm 0.016$ |
| | | True | True | $0.854 \pm 0.007$ | $0.940 \pm 0.001$ | $0.350 \pm 0.017$ |
| | 5 | False | False | $0.874 \pm 0.001$ | $0.940 \pm 0.001$ | $0.397 \pm 0.007$ |
| | | False | False | $0.872 \pm 0.002$ | $0.940 \pm 0.001$ | $0.396 \pm 0.008$ |
| | | True | True | $0.876 \pm 0.002$ | $0.939 \pm 0.001$ | $0.399 \pm 0.006$ |
| | | True | True | $0.872 \pm 0.001$ | $0.939 \pm 0.001$ | $0.393 \pm 0.002$ |
| SoftLabeler | 1 | False | False | $0.832 \pm 0.009$ | $0.941 \pm 0.001$ | $0.305 \pm 0.015$ |
| | | False | False | $0.835 \pm 0.009$ | $0.942 \pm 0.001$ | $0.313 \pm 0.014$ |
| | | True | True | $0.837 \pm 0.004$ | $0.941 \pm 0.001$ | $0.314 \pm 0.007$ |
| | | True | True | $0.839 \pm 0.004$ | $0.941 \pm 0.001$ | $0.322 \pm 0.008$ |
| | 5 | False | False | $0.858 \pm 0.003$ | $0.939 \pm 0.001$ | $0.348 \pm 0.008$ |
| | | False | False | $0.858 \pm 0.004$ | $0.940 \pm 0.001$ | $0.355 \pm 0.010$ |
| | | True | True | $0.861 \pm 0.001$ | $0.940 \pm 0.001$ | $0.351 \pm 0.004$ |
| | | True | True | $0.862 \pm 0.001$ | $0.940 \pm 0.001$ | $0.361 \pm 0.004$ |
| DeepAE | 1 | False | False | $0.863 \pm 0.003$ | $0.942 \pm 0.001$ | $0.376 \pm 0.016$ |
| | | False | False | $0.867 \pm 0.004$ | $0.942 \pm 0.001$ | $0.379 \pm 0.018$ |
| | | True | True | $0.865 \pm 0.002$ | $0.943 \pm 0.001$ | $0.376 \pm 0.009$ |
| | | True | True | $0.869 \pm 0.002$ | $0.942 \pm 0.000$ | $0.381 \pm 0.009$ |
| | 5 | False | False | $0.881 \pm 0.002$ | $0.942 \pm 0.001$ | $0.414 \pm 0.014$ |
| | | False | False | $0.875 \pm 0.001$ | $0.941 \pm 0.001$ | $0.406 \pm 0.009$ |
| | | True | True | $0.882 \pm 0.001$ | $0.941 \pm 0.002$ | $0.415 \pm 0.004$ |
| | | True | True | $0.879 \pm 0.001$ | $0.940 \pm 0.002$ | $0.410 \pm 0.001$ |
| RND | 1 | False | False | $0.859 \pm 0.001$ | $0.942 \pm 0.001$ | $0.354 \pm 0.008$ |
| | | False | False | $0.864 \pm 0.001$ | $0.943 \pm 0.001$ | $0.357 \pm 0.010$ |
| | | True | True | $0.855 \pm 0.004$ | $0.943 \pm 0.001$ | $0.354 \pm 0.012$ |
| | | True | True | $0.860 \pm 0.004$ | $0.942 \pm 0.001$ | $0.359 \pm 0.012$ |
| | 5 | False | False | $0.878 \pm 0.002$ | $0.941 \pm 0.000$ | $0.398 \pm 0.008$ |
| | | False | False | $0.875 \pm 0.001$ | $0.940 \pm 0.000$ | $0.395 \pm 0.006$ |
| | | True | True | $0.875 \pm 0.002$ | $0.941 \pm 0.001$ | $0.389 \pm 0.006$ |
| | | True | True | $0.874 \pm 0.002$ | $0.940 \pm 0.001$ | $0.389 \pm 0.005$ |
| OC | 1 | False | False | $0.861 \pm 0.001$ | $0.938 \pm 0.001$ | $0.371 \pm 0.011$ |
| | | False | False | $0.867 \pm 0.001$ | $0.939 \pm 0.001$ | $0.373 \pm 0.013$ |
| | | True | True | $0.861 \pm 0.003$ | $0.944 \pm 0.001$ | $0.361 \pm 0.014$ |
| | | True | True | $0.866 \pm 0.003$ | $0.943 \pm 0.001$ | $0.364 \pm 0.014$ |
| | 5 | False | False | $0.878 \pm 0.001$ | $0.943 \pm 0.001$ | $0.395 \pm 0.009$ |
| | | False | False | $0.875 \pm 0.001$ | $0.943 \pm 0.001$ | $0.393 \pm 0.008$ |
| | | True | True | $0.876 \pm 0.001$ | $0.941 \pm 0.001$ | $0.390 \pm 0.012$ |
| | | True | True | $0.875 \pm 0.001$ | $0.942 \pm 0.001$ | $0.389 \pm 0.011$ |
| MIMO | 1 | False | False | $0.868 \pm 0.003$ | $0.942 \pm 0.001$ | $0.388 \pm 0.005$ |
| | | False | False | $0.874 \pm 0.003$ | $0.941 \pm 0.001$ | $0.397 \pm 0.005$ |
| | | True | True | $0.867 \pm 0.003$ | $0.941 \pm 0.001$ | $0.392 \pm 0.009$ |
| | | True | True | $0.874 \pm 0.002$ | $0.942 \pm 0.001$ | $0.398 \pm 0.010$ |
| | 5 | False | False | $0.870 \pm 0.003$ | $0.939 \pm 0.001$ | $0.371 \pm 0.014$ |
| | | False | False | $0.864 \pm 0.004$ | $0.939 \pm 0.001$ | $0.366 \pm 0.013$ |
| | | True | True | $0.872 \pm 0.001$ | $0.940 \pm 0.001$ | $0.377 \pm 0.005$ |
| | | True | True | $0.868 \pm 0.001$ | $0.940 \pm 0.002$ | $0.375 \pm 0.004$ |
| MCDropout | 1 | False | False | $0.864 \pm 0.002$ | $0.943 \pm 0.001$ | $0.368 \pm 0.005$ |
| | | False | False | $0.869 \pm 0.002$ | $0.942 \pm 0.001$ | $0.373 \pm 0.007$ |
| | | True | True | $0.865 \pm 0.005$ | $0.942 \pm 0.002$ | $0.380 \pm 0.010$ |
| | | True | True | $0.871 \pm 0.005$ | $0.942 \pm 0.002$ | $0.385 \pm 0.011$ |
| | 5 | False | False | $0.878 \pm 0.001$ | $0.942 \pm 0.001$ | $0.394 \pm 0.007$ |
| | | False | False | $0.874 \pm 0.003$ | $0.942 \pm 0.001$ | $0.391 \pm 0.008$ |
| | | True | True | $0.879 \pm 0.001$ | $0.941 \pm 0.000$ | $0.400 \pm 0.003$ |
| | | True | True | $0.878 \pm 0.002$ | $0.941 \pm 0.000$ | $0.400 \pm 0.003$ |
| DUE | 1 | False | False | $0.553 \pm 0.008$ | $0.950 \pm 0.001$ | $0.038 \pm 0.002$ |
| | | False | False | $0.553 \pm 0.008$ | $0.950 \pm 0.001$ | $0.038 \pm 0.002$ |
| | | True | True | $0.840 \pm 0.000$ | $0.942 \pm 0.000$ | $0.286 \pm 0.000$ |
| | | True | True | $0.836 \pm 0.000$ | $0.942 \pm 0.000$ | $0.285 \pm 0.000$ |

Table 3: Out-domain results for ResNet50 using measure "Entropy"

ETHICS STATEMENT

This work does not involve the release of any new data, or the development of any new algorithm. However, we believe that the development of better uncertainty estimates for our predictive models is at the core of responsible, fair machine learning. As such, we hope that the UIMNET baseline is a stepping stone towards advancing the ability of our systems to say "I don't know". Also, we hope that our manuscript and software aids future research to study uncertainty estimates as components also vulnerable to attacks and with their own set of limitations.

REPRODUCIBILITY

A main focus of this manuscript is reproducibility. To replicate all of our experimental results, obtain our code from the Supplementary Material and follow the instructions at `README.md`. Upon publication, our code will be open-sourced in a GitHub repository, and all of our experimental results will be replicable from a fixed commit hash.

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
