# OpenReview forum: "What classifiers know what they don't know?"
_ICLR.cc/2022/Conference — ICLR 2022 Submitted_

### Official Review · Reviewer_R3aT · 2021-10-31

**Correctness:** 2
**Technical Novelty And Significance:** 1
**Empirical Novelty And Significance:** 2
**Recommendation:** 5
**Confidence:** 4

**Main Review:**

Strengths:
- The paper is well written.
- The benchmarking is solid and very thorough. Many baselines and metrics have been implemented, and results are presented with confidence intervals.

Weaknesses:
- Overall, the major weakness is in the design of this benchmark. For a study claiming to be "the most exhaustive uncertainty estimation benchmark," the only test-time distribution shift considered is the presentation of novel classes. The authors do not appear to justify this design decision anywhere in the paper. The authors make multiple appeals to "providing realistic out-domain data," but there are many, many ways out-domain data can be presented in real world applications. This disjoint class assumption is very strong and is not clear why or when this is representative.
- From reading about the construction of the benchmark, it's not entirely clear why BREEDS wasn't a suitable candidate and why something new had to be created. The authors state that the BREEDS task is to classify super-classes and so the label set remains fixed from train to test - that is true, but could the authors not have just used the same split and kept the leaf node original class labels instead of using the superclass labels?
- The agglomerative clustering method used to construct the splits seems round-about. The ImageNet hierarchy contains a lot of metadata that could have been used to create the splits - do the authors have any comment on why this wasn't used? Also, the result of this split is that the train-test gap is over object classes to animal classes, which is perhaps (a priori) not as realistic of a setting as in the BREEDS dataset, where the train-test gap is between very similar class categories. This design decision was again never justified.

**Summary Of The Paper:**

This paper proposes a new testbed to test the uncertainty of image classifiers based on a split of ImageNet.

**Summary Of The Review:**

In summary, this study has a large number of design flaws. However, given the benchmark, the authors have done a great job in thoroughly benchmarking existing algorithms.

---

### Official Review · Reviewer_tETq · 2021-11-02

**Correctness:** 3
**Technical Novelty And Significance:** 2
**Empirical Novelty And Significance:** 2
**Recommendation:** 3
**Confidence:** 4

**Main Review:**

##########################################################################

Pros:

* Uncertainty estimation is an important topic.

* I commend the authors for their efforts on reproducibility.

##########################################################################

Cons:

* The novelty of the research is low. The data source, algorithms, and metrics are all prior work.

* What value will the community get from the release of ImageNot? The authors have already reported their results. Can the authors outline some additional use cases? Otherwise there is not much point in branding it as a "new dataset".

##########################################################################

Questions during rebuttal period:


* The authors claim in/out domain data should be defined from the same dataset, and that "this provides realistic out-domain data", as opposed to the alternatives (different datasets, augmentations). What is the justification for this claim of greater "realism"?

* How extensible is the software framework to new datasets? What would I have to implement to run the framework on a new dataset?

**Summary Of The Paper:**

This paper proposes the UIMNET benchmark for uncertainty estimation. This benchmark includes (1) "ImageNot", a remix of ImageNet using hierarchical clustering of pairwise distances between features (2) framework for evaluating uncertainty estimation, including a suite of algorithms, metrics, and ablations studies. The authors report empirical results on their dataset using their suite, and make a number of recommendations based on these findings. Reproducible software is also provided.

**Summary Of The Review:**

New empirical benchmark with reproducible software but with unclear research novelty and value to community.

---

### Official Review · Reviewer_tta3 · 2021-11-03

**Correctness:** 3
**Technical Novelty And Significance:** 2
**Empirical Novelty And Significance:** 3
**Recommendation:** 6
**Confidence:** 4

**Main Review:**

For Strengths, the paper provides a solid test-bed for evaluating the uncertainty estimation for deep image classifiers. Also, the paper is well-written and both the data construction and results part are clearly described.
The main weakness of the paper is that the uncertainty measurements in Section 4 lack the SOTA uncertainty estimation methods. For example, no method in Section 4 measures the epistemic uncertainty such as deep ensemble/dropout and aleatoric uncertainty [1] explicitly. Also, recent dirichlet-distribution-based approaches [2][3] are not included to measure the evidence-based uncertainty.

[1] Kendall, Alex, and Yarin Gal. "What uncertainties do we need in bayesian deep learning for computer vision?." arXiv preprint arXiv:1703.04977 (2017).
[2] Sensoy, Murat, Lance Kaplan, and Melih Kandemir. "Evidential deep learning to quantify classification uncertainty." arXiv preprint arXiv:1806.01768 (2018).
[3] Charpentier, Bertrand, Daniel Zügner, and Stephan Günnemann. "Posterior network: Uncertainty estimation without ood samples via density-based pseudo-counts." arXiv preprint arXiv:2006.09239 (2020).


**Summary Of The Paper:**

The paper introduce a benchmark UIMNET to evaluate predictive uncertainty estimates for deep image clasifiers. The authors provides implements of ten state-of-the-art algorithms and six uncertainty measures with four in-domain metrics and three out-domain metrics.

**Summary Of The Review:**

Overall, I think the paper is well-written and the benchmark is valuble as testbed for uncertainty estimation. But the uncertainty measurement part lacks some important uncertainty estimation methods, which are highly encouraged to be included in the paper.

---

### Decision · Program_Chairs · 2022-01-20

**Decision:**

Reject

**Comment:**

This paper introduces an ImageNet-scale benchmark UIMNET for uncertainty estimation of deep image classifiers and evaluates prior works under the proposed benchmark. Two reviewers suggest reject, and one reviewer does acceptance. In the discussion period, the authors did not provide any response for many concerns of reviewers, e.g., weak baselines, weak novelty, and lack of justification for the current design. Hence, given the current status, AC recommends reject.